# Analysis of *Centranthera grandiflora* Benth Transcriptome Explores Genes of Catalpol, Acteoside and Azafrin Biosynthesis

**DOI:** 10.3390/ijms20236034

**Published:** 2019-11-29

**Authors:** Xiaodong Zhang, Caixia Li, Lianchun Wang, Yahong Fei, Wensheng Qin

**Affiliations:** 1College of Chemistry Biology and Environment, Yuxi Normal University, Yuxi 653100, China; zxd95@yxnu.edu.cn (X.Z.); lcx@yxnu.edu.cn (C.L.); wanglianchun@yxnu.edu.cn (L.W.); 2Food and Bioengineering College, Xuchang University, Xuchang 461000, China; 3Yuxi Flyingbear Agricultural Development Company Limited, Yuxi 653100, China; feiyahong@gmail.com; 4Department of Biology, Lakehead University, Thunder Bay, ON P7B 5E1, Canada

**Keywords:** *Centranthera grandiflora* Benth, transcriptome, catalpol biosynthesis, acteoside biosynthesis, azafrin biosynthesis

## Abstract

Cardiovascular diseases (CVDs) are a major cause of health loss in the world. Prevention and treatment of this disease by traditional Chinese medicine is a promising method. *Centranthera grandiflora* Benth is a high-value medicinal herb in the prevention and treatment of CVDs; its main medicinal components include iridoid glycosides, phenylethanoid glycosides, and azafrin in roots. However, biosynthetic pathways of these components and their regulatory mechanisms are unknown. Furthermore, there are no genomic resources of this herb. In this article, we provide sequence and transcript abundance data for the root, stem, and leaf transcriptome of *C. grandiflora* Benth obtained by the Illumina Hiseq2000. More than 438 million clean reads were obtained from root, stem, and leaf libraries, which produced 153,198 unigenes. Based on databases annotation, a total of 557, 213, and 161 unigenes were annotated to catalpol, acteoside, and azafrin biosynthetic pathways, respectively. Differentially expressed gene analysis identified 14,875 unigenes differentially enriched between leaf and root with 8,054 upregulated genes and 6,821 downregulated genes. Candidate MYB transcription factors involved in catalpol, acteoside, and azafrin biosynthesis were also predicated. This work is the first transcriptome analysis in *C. grandiflora* Benth which will aid the deciphering of biosynthesis pathways and regulatory mechanisms of active components.

## 1. Introduction

Cardiovascular diseases (CVDs) are an important reason for death in the world which hinder sustainable development of human beings [1]. In China, CVDs were also the leading cause of death due to lifestyle changes, urbanization, and the accelerated process of aging, and the figures have exceeded 42% of all deaths in both rural and urban regions, which was much higher than deaths caused by cancer or any other diseases in 2014 [2]. Traditional Chinese medicine has been used for more than 2000 years and has displayed the explicit role in preventing and treating CVDs, although the detailed pharmacological mechanisms have seldomly been clarified [3]. *Centranthera grandiflora* Benth, also known as broad bean *Ganoderma lucidum*, wild broad bean root, Huaxuedan, Golden Cat’s Head, and Xiaohongyao, is a medicinal plant widely used for preventing and treating CVDs among Miao Nationality of Yunnan in China. In taxonomy, it belongs to the Centranthera, Scrophulariaceae family. Distinguished as a rare and endangered medicinal plant, *C. grandiflora* Benth usually grows well with *Cyperus rotundus* and is mainly distributed in Yunnan, Guizhou, and Guangxi in China as well as parts of India, Myanmar, and Vietnam [4,5,6,7]. Its roots possess many functions, such as to promote blood circulation, to regulate menstruation, to dispel blood stasis, and to relieve pain, and has known coagulation, antibacterial, and anticancer properties [6,8,9,10]. Therefore, it is mainly used to treat amenorrhea, dysmenorrhea, metrorrhagia, fall-related injuries, rheumatic bone pain, traumatic hemorrhage, and cardiovascular and cerebrovascular diseases [6,8,9,10].

So far, studies on this herb have mainly focused on the isolation and identification of its chemical constituents and pharmacological effects, while the discovery of genes related to biosynthesis of active secondary metabolites has not been reported. Azafrin and D-mannitol were first isolated and identified from the roots of *C. grandiflora* Benth in 1984 [8]. Then, aeginetin and azalea were isolated from the roots of *C. grandiflora* Benth, and their coagulation, antimicrobial, and anticancer functions were verified in 2012 [10]. In the same year, nine iridoid glycosides including aucubin, mussaenoside, 8-epiloganin, 8-epiloganic acid, mussaenosidic acid, catalpol, gardoside methyl ester, geniposidic acid, and 6-O-methylaucubin were isolated from roots of *C. grandiflora* Benth [6]. In 2014, another 17 compounds, including six new ones: centrantheroside A to E and neomelasmoside; phenylethanoid glycosides: plantainoside A, calceolarioside A, acteoside, and isoacteoside; monoterpenoid glycosides: melasmoside and rehmaionoside C; Di-O-methylcrenatin; azafrin; β-sitosterol; mannitol; and β-daucosterol were isolated from *C. grandiflora* Benth roots [5,7]. Studies have shown that iridoid glycosides, phenylethanoid glycosides, and azafrin are the main substance bases for their pharmacodynamics [5,7]. In 2017, tissue culture of *C. grandiflora* Benth was also successfully developed [11].

At present, *C. grandiflora* Benth roots sold in public markets are mainly collected from wild resources, while its artificial cultivation has just started [5]. So far, the cost of annual *C. grandiflora* Benth planting is about $0.13 million per hectare and the worth of annual yield is about $0.64 million per hectare [5]. Therefore, to explore the biosynthetic pathways and regulatory mechanisms of the main active ingredients of *C. grandiflora* Benth will lay a scientific foundation for breeding new varieties of this herb and for producing its medicinal chemical constituents by synthetic biology.

Iridoid glycosides belong to monoterpenoids, and their biosynthesis in plants can be divided into three stages. The first stage is precursor formation, which includes the plastidial 2-C-methyl-D-erythritol-4-phosphate (MEP) pathway and the cytoplasmic mevalonate (MVA) pathway to produce isopentenyl diphosphate (IPP) and dimethylallyl diphosphate (DMAPP) [12,13]. The second stage is the formation of a carbon skeleton structure [13,14,15,16]. The third stage is the post-modification of terpenoids: hydroxylation, methylation, isomerization, demethylation, glycosylation, etc. [16]. So far, most of the biosynthesis pathways of iridoid glycosides remain unclear. However, the complete catalpol biosynthetic pathway was first elicited in *Picrorhiza kurroa* [17], and it was partially decoded in *Rehmannia glutinosa* [18]. In *P. kurroa*, the catalpol biosynthetic pathway contains 29 steps including 14 steps for the MEP and MVA pathways and 15 steps for the iridoid pathway [17]. As the MEP and MVA pathways has been widely and intensively studied and they are conserved in plants [19], here, we mainly focused on the iridoid pathway. So far, two iridoid pathways including secoiridoid pathway (Route I) and decarboxylated iridoid pathway (Route II) have been reported, and the early enzymatic steps containing geranyl diphosphate synthase (GPPS), geraniol synthase (GES), geraniol 10-hydroxylase (G10H), 8-hydroxygeraniol oxidoreductase (8HGO), iridoid synthase (IS), iridoid oxidase (IO), and UDP-glucosyltransferase (UGT) are common to both pathways [20], has been verified in *Catharanthus roseus* and *P. kurroa* [14,21,22], and proposed in *Gardenia jasminoides* [23,24]. The remaining steps were first deduced by chemical intermediates [20], and then the corresponding enzymes were predicted and discovered by transcriptome analysis [17]. In *P. kurroa*, another seven enzymes containing aldehyde dehydrogenase (ALD), flavanone 3-dioxygenase/hydoxylase (F3D), 2-hydroxyisoflavanone dehydratase (2FHD), deacetoxycephalosporin-C hydroxylase (DCH), uroporphyrinogen decarboxylase/UDP-glucuronic acid decarboxylase (UPD/UGD), and squalene monooxygenase (SQM) have been proposed to catalyze the remaining seven steps in catalpol biosynthesis [17].

Acteoside, belonging to phenylethanoid glycosides, is composed of two parts: caffeoyl CoA and hydroxytyrosol glucoside [25]. Feeding and inhibition experiments showed that hydroxytyrosol glucoside moiety is derived from tyrosine while caffeoyl CoA moiety is derived from phenylalanine via the cinnamate pathway and that both tyrosine and phenylalanine come from the shikimate pathway [26,27]. In *Ole europae* and *R*. *glutinosa*, phenylalanine is converted into caffeoyl CoA via four enzymes including phenylalanine ammonia-lyase (PAL), cinnamate-4-hydroxylase (C4H), coumarate-3-hydroxylase (C3H), and 4-coumarate-CoA ligase (4CL) [18,25,28]. Simultaneously, tyrosine is transformed into hydroxytyrosol glucoside through two alternative pathways: one is via *L*-dopa, dopamine, and hydroxytyrosol with the enzymes polyphenol oxidase (PPO), tyrosine decarboxylase (TDC), copper-containing amine oxidase (CuAO), alcohol dehydrogenase (ADH), and UGT; the other is via tyramine, tyrosol, and salidroside with the enzymes TDC, CuAO, ADH, UGT, and PPO [18,25,28]. Finally, caffeoyl CoA and hydroxytyrosol glucoside can be converted into acteoside by Shikimate O-hydroxycinnamoyltransferase (HCT) and UGT [18,25]. Recent studies have verified that acteoside possesses pharmacological properties: antioxidant, anti-inflammatory, antidepressant, antitumor, antidiabetes, and hepatoprotection [29,30,31,32].

Azafrin, belonging to carotenoid derivative, is one of the most abundant active ingredients in *C. grandiflora* Benth roots and plays an important role in myocardial protection [33]. Carotenoids are ubiquitous pigments in plants, and they confer plants with bright yellows, oranges, and reds [34]. In higher plants, carotenoids are synthesized through isoprene-like pathways in plastids, including condensation, dehydrogenation, cyclization, hydroxylation, and epoxidation reactions, while lycopene acts as an important branch point of both synthesis of α-carotene and β-carotene [35]. In the α-carotene pathway, α-carotene is synthesized by lycopene ε-cyclase (LCY-ε) and lycopene β-cyclase (LCY-β) and is then converted to lutein by ε-hydroxylase (LUT1) and β-cyclohexylase (LUT5) [17,18]. In the β-carotene pathway, LCY-β catalyzes the synthesis of β-carotene, which can be converted into strigolactone, astaxanthin, capsanthin, capsorubin, and violaxanthin under the catalysis of different enzymes, while violaxanthin can be further converted into abscisic acid [36,37]. However, studies have shown that azafrin is an apocarotenoid which is generated by cleavage of carotenoids at the C9′–C10′ [38,39]. In the strigolactone pathway, β-carotene is converted into carlactone through 9-cis-carotene, 10′-apo-β-carotenal by enzymes DWARF27, carotenoid cleavage dioxygenase 7 (CCD7), and CCD8 [39]. The intermediate product 10′-apo-β-carotenal is very similar to azafrin in structure except one terminal carboxyl group and two hydroxyl groups. Therefore, the hypothesis that azafrin is synthesized via 10′-apo-β-carotenal is proposed in this article.

Thus, the aim of this research is to characterize globally for the first time the transcriptomes of the root, stem, and leaf of *C. grandiflora* Benth using the Illumina Hiseq2000. To explore the genes involved in the catalpol, acteoside, and azafrin biosynthesis pathways and regulatory mechanisms, transcripts from leaves, stems, and roots of *C. grandiflora* Benth were screened out, quantified, and annotated. The results obtained here will facilitate further molecular studies in *C. grandiflora* Benth.

## 2. Results

### 2.1. Sequencing and Assembly

To figure out which genes are involved in the biosynthesis of active components in *C. grandiflora* Benth, nine sequencing libraries including roots (C_R1, C_R2, and C_R3), stems (C_S1, C_S2, and C_S3), and leaves (C_L1, C_L2, and C_L3) were prepared and sequenced with the Illumina Hiseq2000 platform. As a result, more than 45 million clean reads per library were obtained after cleaning and quality examination. Quality assessments of the sequencing data are shown in Table 1. The error rate of all libraries was 0.03%, while Q20 and Q30 were over 94.99% and 88.32%, respectively, indicating that these data are suitable for further analysis. The raw data from the nine libraries have been deposited into the Short Reads Archive (SRA) database under the accession numbers: SRX6654843–SRX6654851.

The clean reads were combined and assembled by Trinity with min_kmer_cov set to 2 and all other default parameters [40]. Assembled sequences were subjected to cluster using the Trinity algorithm. As a result, 153,300 contigs were clustered into 173,851 trinity components. Each Trinity component contained a set of transcripts derived from the same gene, and a unigene was designated as the longest transcript in each trinity component. A total of 173,851 transcripts and 153,198 genes were assembled, with 69,421 (39.93%) transcripts and 69,419 (45.31%) genes being over 2 Kb in length (Figure 1). The average length of transcripts and genes were 1895 bp and 2115 bp, respectively (Table 2), and the N50 for transcripts and genes were 2902 and 2936 bp, respectively (Table 2). 

### 2.2. Gene Function Annotation and Classification

All the 153,198 assembled putative unigenes were aligned using the BLAST (Basic Local Alignment Search Tool) program against the seven classic databases including NR (nonredundant protein sequences), NT (Nucleotide collection), PFAM (Protein family), SwissProt, KOG (euKaryotic Orthologous Groups), KEGG (Kyoto Encyclopedia of Genes and Genomes), and GO (Gene Ontology) databases with *e*-value cutoffs of 10^−5^, 10^−5^, 10^−2^, 10^−5^, 10^−3^, 10^−10^, and 10^−6^, respectively. A total of 26,652 unigenes (17.39%) were annotated to the above seven databases in common, while 132,896 unigenes (86.74%) were annotated in at least one database (Table 3). Among them, 127,767 unigenes (83.39%) showed high similarity with sequences in the NR database (*e*-value = 10^−5^), 96,216 unigenes (62.80%) matched to protein sequences in NT, and 103,257 unigenes (67.40%) showed homology with known genes in SwissProt. The detailed results are shown in Table 3 and Appendix A. Based on the top-hit species distribution of the homology results against NR database, the highest matches were genes from *Sesamum indicum* (43.77%), followed by *Handroanthus impetiginosus* (22.58%) and *Erythranthe guttata* (13.07%) (Figure 2).

In coding sequence prediction analysis, unigenes were aligned first to the NR and then Swissprot database. If aligned, ORF (open reading frame) information of transcripts was extracted from the alignment results and the sequence of the coding region was translated into amino acid sequences according to the standard codon table. If failed, ESTSCAN (Expression Sequence Tag Scan) software was adopted to predict the ORF of the unigenes. As a result, a total of 157,392 peptides were predicted by BLASTx and the peptide length mainly ranged from 38 to 1059 (Figure 3a) while 35,339 peptides were predicted by ESTSCAN and the peptide length was from 15 to 1092 (Figure 3b).

To figure out the biological processes that our unigenes are involved in as well as their molecular functions and the cellular environments they reside in, all unigenes were searched against the GO database with software BLAST2GO. Out of 153,198 unigenes, 98,364 (64.20%) were successfully annotated and classified into three GO categories—biological process (BP), cellular component (CC), and molecular function (MF)—and then assigned to 55 functional groups (Figure 4). As shown in Figure 4, assignments which fell under BP (273,598, 47.25%) ranked the highest, followed by CC (175,650, 30.34%) and MF (129,742, 22.41%). Similar to *R. glutinosa* [41] and adventitious roots in *Panax ginseng* [42], “cellular process” (60,529, 61.54%) and “metabolic process” (56,468, 57.41%) were the two most representative subcategories in the BP category, which suggested that lots of important cellular processes and metabolic activities took place in *C. grandiflora* benth. Unlike adventitious roots in *P*. *ginseng* [42], although unigenes related to “cell” (33,946, 34.51%) and “cell part” (33,946, 34.51%) were dominant in the CC category, the percentages in *C. grandiflora* Benth were far less than in *P*. *ginseng*, which implied that many tissues and organs in *C. grandiflora* Benth were in construction at a slow speed. In the MF category, the majority of unigenes were involved in “binding” (59,157, 60.14%) and “catalytic activity” (49,089, 49.91%) in *C. grandiflora* Benth, and this was somewhat similar to *R. glutinosa* [41], in which unigenes annotated into “binding” were about 20% more than “catalytic activity”, indicating that more catalytic reactions may occur in the form of protein complexes. 

KOG refers to clusters of orthologous groups from different eukaryotic species, and genes from the same ortholog are assumed to have the same function. To further classify our unigenes, KOG annotation was performed with software diamond. A total of 44,170 unigenes were classified into 26 KOG groups (Figure 5), where the “posttranslational modification, protein turnover, and chaperon” (5516, 12.49%) category accounted for the most frequent group, “general function prediction only” (5445, 12.33%) was the second largest group, and “translation, ribosomal structure, and biogenesis” (4404, 9.97%) and “intracellular trafficking, secretion, and vesicular transport” (3504, 7.93%) were tied for the third largest. In addition, 773 unigenes were assigned to “secondary metabolites biosynthesis, transport, and catabolism”, implying that catalpol, acteoside, and carotenoid biosynthesis may take place in *C. grandiflora* Benth.

KEGG is a database in which gene products and compounds of the cellular metabolic pathways and the functions of these gene products were systematically analyzed. To figure out the active pathways in growing *C. grandiflora* Benth, KEGG annotation of our unigenes were performed with KAAS (KEGG Automatic Annotation Server). A total of 57,190 (37.33%) unigenes were annotated into five categories (level 1; cellular processes, environmental information processing, genetic information processing, metabolism, and organismal systems), 19 subcategories (level 2, Figure 6), and 130 pathways (level 3). Similar to *R. glutinosa* [41], the five pathways with the largest number of genes were “carbohydrate metabolism” (4996, 8.74%), “overview” (3455, 6.04%), “amino acid metabolism”, “lipid metabolism”, and “energy metabolism” in the metabolism category, indicating that primary metabolism was very important to the growth of *C. grandiflora* Benth. In the category of genetic information processing, the two pathways with the largest number of genes were “translation” (5427, 9.49%) and “folding, sorting, and degradation” (4178, 7.31%), indicating that protein biosynthesis and processing were more active in *C. grandiflora* Benth (Figure 6). The numbers of unigenes for “amino acid metabolism”, “metabolism of terpenoids and polyketides”, and “biosynthesis of other secondary metabolites” were 2927, 1164, and 882, respectively. These results indicate that the amino acid pathway and terpenoid pathways were active in growing *C. grandiflora* Benth and that the corresponding genes would be good candidate genes for catalpol, acteoside, and carotenoid biosynthesis.

### 2.3. Identification of Differentially Expressed Genes (DEGs), GO, and KEGG Enrichment Analysis

Gene expression level which is transformed from read counts using RSEM (RNA-Seq by Expectation-Maximization) software was analyzed with the FPKM (expected number of Fragments Per Kilobase of transcript sequence per Millions base pairs sequenced) method [43]. According to the criteria *p* < 0.05 and log_2_(FoldChange) > 1, 14,875 genes (9.71% of all genes) were identified as significant DEGs between leaves and roots, which comprised 8054 upregulated genes (54.14%) and 6821 downregulated genes (45.86%) in leaves (Figure 7a, Appendix A). There were 4126 genes (2.69% of all genes) identified as significant DEGs between leaves and stems, which comprised 2251 upregulated genes (54.56%) and 1875 downregulated genes (45.44%) in leaves (Figure 7b, Appendix A). A total of 9115 genes (5.95% of all genes) were identified as significant DEGs between stems and roots, which comprised 5290 upregulated genes (58.04%) and 3825 downregulated genes (41.96%) in stems (Figure 7c, Appendix A). Using a Venn diagram, we compared these three data sets from different comparison groups (C_L vs. C_R, C_S vs. C_R, and C_L vs. C_S). In all three comparison groups, 829 DEGs were identified as being in common (Figure 7d). Specifically, 4839 DEGs were identified in both “C_L vs. C_S” and “C_S vs. C_R” comparisons; 1918 DEGs were identified in both “C_L vs. C_R” and “C_L vs. C_S” comparisons; and 551 DEGs were identified in both “C_L vs. C_S” and “C_S vs. C_R” comparisons (Figure 7d).

GO and KEGG enrichment analysis on all DEGs were performed to find the enriched pathways. The GO enrichment is shown in Appendix A. In the KEGG enrichment analysis, the top two enriched pathways were flavonoid biosynthesis with 37 DEGs and 68 background unigenes, and flavone and flavonol biosynthesis with 10 DEGs and 14 background unigenes in the C_L vs. C_R comparison (Figure 8a, Appendix A). In the C_L vs. C_S comparison, the top two pathways were flavone and flavonol biosynthesis with 5 DEGs and 14 background unigenes, and the stilbenoid, diarylheptanoid, and gingerol biosynthesis with 11 DEGs and 48 background unigenes (Figure 8b, Appendix A). Finally, in the L_S vs. L_R comparison, the top two pathways were stilbenoid, diarylheptanoid, and gingerol biosynthesis with 15 DEGs and 48 background unigenes, and flavonoid biosynthesis with 3 DEGs and 11 background unigenes (Figure 8c, Appendix A). Notably, pathways for both phenylpropanoid biosynthesis and carotenoid biosynthesis were enriched in all three comparisons. The top 20 KEGG enrichment pathways are shown in Figure 8.

### 2.4. Biosynthetic Genes of the Terpenoid Backbone and Catalpol in C. grandiflora Benth

Terpenoids are produced from the universal precursor IPP (a five-carbon building material) and its isomer DMAPP [44]. In plants, IPP is synthesized via the cytoplasmic MVA pathway from acetyl-CoA and through the plastidial MEP pathway from glyceraldehyde 3-phosphate and pyruvate; IPP isomerase (IDI) catalyzes the interconversion between IPP and DMAPP [44] (Figure 9a). When examining the annotation of unigenes against the KEGG database, 239 unigenes were assigned to the terpenoid backbone biosynthesis pathway, including 74 unigenes encoding 6 enzymes in the MVA pathway and 165 unigenes encoding 8 enzymes in the MEP pathway (Table 4). Among these genes, the largest number is *DXS* (67), followed by *IDI* (28), *AACT* (24), and *CMK* (24), and the lowest number is *MCS* (1). Transcriptome profiling data showed that the MEP pathway is more active in leaves, while the MVA pathway is more active in stems due to the high expression levels of corresponding pathway genes (Figure 9b).

Catalpol, belonging to iridoid glucoside, is usually found in Scrophulariaceae plants [6,45,46], and iridoid glucosides are derived from MEP and MVA pathways [41]. Based on feeding experiments with isotope labeling and transcriptome analysis, the draft biosynthesis pathway of catalpol was first proposed in *R. glutinosa* in 2012 [47,48]. Then, the complete pathway of catalpol was clarified for the first time in *P. kurroa* in 2015 [17].

According to the KEGG and Swissprot annotation, a total of 368 unigenes were assigned to the catalpol biosynthetic pathway, with 60 unigenes upregulated and 39 unigenes downregulated in leaf vs. root (Figure 9a and Table 4). The unigenes encoding 13 enzymes involved in catalpol biosynthesis are listed in Table 4. Among these genes, the largest number was *ALDH* (76), followed by *8HGO* (53), *IO* (44), *GPPS* (32), and UGT (30), and the lowest number was *F3D* (2) (Table 4). Like the terpenoid backbone pathway, most genes in the catalpol biosynthesis pathway possess at least two unigenes, displaying the redundancy of the plant genes and adding the difficulty of deciphering the pathway (Table 4). In our transcriptome, unigenes of catalpol pathways are more abundant in leaves, as revealed by much higher expression level of *GES*, *G10H*, *IS*, *IO*, and *F3D* in leaves than in roots (Figure 9b, Table 4). It is worth noting that there were only two *F3D* genes in our transcriptome and that it was only expressed in leaves and stems, but not roots, which indicated that the catalpol biosynthesis was active in aboveground growth at this developmental stage (Figure 9a, Table 4). Therefore, while it is the roots of *C. grandiflora* Benth that are used as medicinal materials, our results imply that the catalpol is first synthesized in the leaves and then transported and stored in the roots. Furthermore, *DCH* gene functioning in the conversion of deoxygeniposidic acid to geniposidic acid was not found in our transcriptome, which may be a result of low expression or low homology with the known *DCH* genes. 

### 2.5. Biosynthetic Genes of Acteoside in C. grandiflora Benth

Studies have shown that acteoside is widely distributed in more than 150 plant species and has medicinal properties including antioxidant, anti-inflammation, anti-nephritis, cell regulation, hepatoprotection, immunoregulation, and neuroprotection [49]. Upstream regions of the acteoside biosynthetic pathway, including the phenylalanine-derived pathway and tyrosine-derived pathway, was first clarified in *Olea europaea* using feeding experiments, while the downstream is largely unknown [26]. The downstream region was partially deciphered with elicitor inducing and transcriptome sequencing in *R. glutinosa* [18,28].

Based on the KEGG annotation in this study, a total of 213 unigenes were assigned to the acteoside biosynthetic pathway, with 40 unigenes significantly upregulated and 16 unigenes significantly downregulated in leaves vs. roots (Figure 10a). The unigenes encoding key enzymes involved in acteoside biosynthesis are listed in Table 5. Among these genes, the largest number was *CuAO* (53), followed by *4CL* (35), *ADH* (26), and *UGT* (22), and the lowest number was *HCT* (6) (Table 5). In the DEGs analysis, four genes including *PAL*, *C4H*, *C3H*, and *4CL* were upregulated in leaves and stems compared with roots (Figure 10b, Table 5), which implies that the phenylalanine-derived pathway is active in aerial parts.

### 2.6. Biosynthetic Genes of Azafrin in C. grandiflora Benth

Recent studies have shown that azafrin can significantly improve myocardial contractile function during myocardial ischemia via activation of the Nrf2-ARE (Nuclear factor-erythroid 2-related factor-Antioxidant Response Element) pathway in rats [7,33]. So far, biosynthesis and chemical synthesis pathways of azafrin are still unknown. Azafrin is a derivative of carotenoid, a tetraterpenoid compound [38]. In higher plants, carotenoids are manufactured in plastid with IPP generated by the MEP pathway [50]. The putative carotenoid biosynthesis pathway, including the MEP part, lutein branch, strigolactone branch, capsanthin/capsorubin branch, abscisic acid branch, and without the azafrin pathway, has been established in plants (Figure 11a) [35,51]. However, studies have shown that the substrate β-carotene can be directly converted into 10‘-apo-β-carotenal and ionone by β-carotene-9′,10′-oxygenase (BCO2) in non-plants or can be indirectly converted into 10′-apo-β-carotenal and ionone by DWARF27 and carotenoid cleavage dioxygenases 7 (CCD7) in plants [39,52]. The differences between azafrin and 10′-apo-β-carotenal are one terminal carboxyl group and two hydroxyl groups in the cyclohexane skeleton. From the aspect of biochemistry, acetaldehyde dehydrogenase (ALDH) can transform aldehyde into carboxylic acid and the cytochrome P450 monooxygenases (CYP450s) are capable of inserting oxygen atoms into inert hydrophobic molecules to make them more hydrophilic [53]. There is also a report that post-modification of terpenoid derivatives is mostly initiated by oxidation and that most of them are catalyzed by CYP450s and then other post-modification reactions are carried out on the basis of oxidation products [54]. Then, we hypothesize that azafrin is produced in two continuous steps: 10′-apo-β-carotenal is first oxidized by ALDH and then is hydrolyzed by CYP450. The detailed biosynthetic pathway of azafrin is shown in Figure 11a.

Based on the KEGG annotation and NR annotation in this study, a total of 356 unigenes were correlated with the carotenoid biosynthesis, of which 161 unigenes were assigned to the azafrin biosynthetic pathway with 20 unigenes upregulated and 33 unigenes downregulated in leaves vs. roots (Table 6). For the MEP portion, it was active in leaves, stems, and roots in general as it is known to provide the universal precursor for the terpenoids (Figure 11b). For the lutein pathway, it was more active in leaves and stems while somewhat inactive in roots due to the low expressions of *LUT5* and *LUT1* genes (Figure 11b). For the azafrin and strigolactone branch, it was slightly active in stem, as neither *DWARF27* and *CCD7* were expressed in leaves nor *CCD7* were expressed in roots (Figure 11b). For the capsanthin/capsorubin and abscisic acid branch, it is more active in leaves and stems and somewhat blocked by the low expression of the *LUT5* gene (Figure 11b). What should be noted here is that there is only one gene encoding the CCD7 enzyme, which may be a rate-limiting enzyme (Table 6).

### 2.7. Identification of Transcription Factors (TFs)

Transcription factors can activate or inhibit the expression of functional genes in the biosynthetic pathway of plant metabolites, thereby effectively regulating the synthesis and accumulation of secondary metabolites. According to gene sequence alignment to the PFAM database, referring to the Hidden Markov Model files of various TFs, the HMMER3.0 software was used to search the transcriptome database of *C. grandiflora* Benth. The results showed that, in our transcriptome, 4888 unigenes were annotated as TFs belonging to 78 categories. The top three TFs with the largest numbers were MYB (avian myeloblastosis viral oncogene homolog, 356, accounting for 7.28%), WRKY (WRKY domain-containing protein, 301, accounting for 6.16%), and orphans (234, accounting for 4.79%), followed by HB (homeobox, 223, accounting for 4.56%), C3H (Cys3His zinc finger domain-containing protein, 209, accounting for 4.28%), and bHLH (basic Helix-Loop-Helix, 201, accounting for 4.11%) (Figure 12, Table 7, and Appendix A). There were also TFs ERF and bZIP (basic region-leucine zipper, Table 7). Among these TFs, most were expressed in both root and leaf tissues, with 121 and 132 showing significantly upregulation and downregulation in leaves, respectively (Table 7).

Studies have also shown that the active components of medicinal plants are regulated by many TFs and that the number of genes regulated by a specific TF varies widely. There may be even crosstalk between regulations. In *Artemisia annua,* only AaHD1 (Homeodomain-leucine zipper) and AaGSW1 (Glandular trichome-Specific WRKY 1) can activate transcription of the *CYP71AV1* gene [55,56]; AaWRKY1, AabHLH1, and ERF (Ethylene Response Factor) TFs including AaTAR1 (Trichome and Artemisinin Regulator 1), AaERF1, AaERF2, and AaORA (Octadecanoid-derivative Responsive *Apetala*2 domain) can activate the transcription of both the amorpha-4,11-diene synthase (*ADS*) gene and the *CYP71AV1* gene and then facilitates artemisinin biosynthesis [57,58,59,60,61]; AabZIP1 is responsible for the activation of the *ADS*, *CYP71AV1*, and *AaGSW1* genes [62], while AaMYC2 is responsible for the *CYP71AV1*, *DBR2* (*Double-Bond Reductase 2*), and *AaGSW1* genes [63,64]; and AaMYB2 may regulate the *ADS*, *CYP71AV1*, *DBR2*, and *ALDH1* (*Aldehyde Dehydrogenase 1*) genes [65]. All the abovementioned TFs were found in our transcriptome (Table 7).

In order to figure out which TFs are involved in catalpol, acteoside, and carotenoid biosynthesis in *C. grandiflora* Benth, MYB TFs of which the log_2_(FC) > 4 were selected for performing phylogenetic analysis with 168 MYBs from *Arabidopsis thaliana*. As a result, a total of 28 MYBs, including 16 upregulated and 12 downregulated, were screened out in leaf vs. root. In kiwifruit, *AdMYB7*, *AdMYB8*, *AdMYBR2*, and *AdMYBR3* play important roles in regulating carotenoid accumulation in tissues through transcriptional activation of metabolic pathway genes [35,66]. In our analysis, CgMYB18, CgMYB26, CgMYB19, and AdMYB7 were all clustered into the S20 subgroup, while AdMYB8 was near the S20; CgMYB15, CgMYB4, CgMYB8, CgMYB13, AdMYBR2, and AdMYBR3 were in the same clade; and *CgMYB18*, *CgMYB26*, *CgMYB19*, *CgMYB15*, *CgMYB4*, *CgMYB8*, and *CgMYB13* were all upregulated in the root, which indicates that they may regulate carotenoid biosynthesis in the roots of *C. grandiflora* Benth. (Figure 13a). In *A. annua*, overexpression of *AaMYB1* exclusively in trichomes or in whole plants both increased the expression of the *FDS* (*farnesyl diphosphate synthase*), *ADS*, *CYP71AV1*, *DBR2,* and *ALDH1* genes and increased the accumulation of artemisinin [67]. CgMYB9 and AaMYB1 were clustered into S13 subgroup and *CgMYB9* was upregulated in leaves, which showed that it may be a candidate regulatory gene in catalpol and carotenoid biosynthesis [66]. Overexpression of *AtPAP1* (*Production of Anthocyanin Pigment1*) in rose plants enhanced production of phenylpropanoid and terpenoid scent compounds by transcriptional activation of their respective pathway genes [68], and AtPAP1, AtMYB90 (AtPAP2), AtMYB113, and AtMYB114 all regulated anthocyanin biosynthesis [69]. In our data, CgMYB1, CgMYB2, CgMYB6, AtPAP1, and AtMYB90 were all in the S6 subgroup and *CgMYB1*, *CgMYB2*, and *CgMYB6* were all significantly upregulated in the leaves (Figure 13b), suggesting that they are candidate regulatory genes in catalpol, acteoside, and carotenoid biosynthesis. 

### 2.8. Expression Correlation Analysis of Selected Genes

To verify our transcriptome results, five terpenoid-related genes including three upregulated genes (*MCS*, *GES*, and *IS*) and two downregulated genes (*8HGO* and *HMGR2*) in leaf vs. root were selected for correlation analysis. All the selected genes possessed the same expression trend although the expression levels varied between RNA-Seq and qRT-PCR, especially for the *8HGO* and *HMGR2* genes (Figure 14a). The overall correlation coefficient was about 0.84, which indicates that our transcriptome is valid (Figure 14b).

## 3. Discussion

Cardiovascular diseases (CVDs) remain a major cause of health loss for all regions of the world in the past 25 years [70]. In China, the incidence of CVDs is continuously rising and will keep an upward trend in the next decade [2]. Therefore, to find the herbs with effective treatment of CVDs is imminent. *C. grandiflora* Benth is one of the most precious herbs in the area of Miao Nationality in Yunnan, China. It is widely used in folk medicine because of its multifunctional medicinal values, especially in the aspect of the prevention and treatment of CVDs [7]. Although it has been collected in the Chinese Materia Medica, up to now, it has not been included in the Chinese Pharmacopoeia because of limited researches [9]. The current situation is high market prices and overexploitation of wild resources, which has not only prevented the herbal medicine from being widely used but has destroyed species diversity. Synthetic biology will provide solutions for the abovementioned problems through the biosynthetic pathway elucidations of the main pharmacodynamic components.

So far, de novo transcriptome analysis is an important method in gene discovery of biosynthesis pathways, especially for species without reference genomes [13]. In this research, the transcriptomes of three tissues with three biological repeats were sequenced by illumine Hiseq2000, and 438,112,930 clean reads were assembled into 173,851 transcripts and 153,198 unigenes. This suggests that one gene may have different transcripts which may come from variable splicing, alleles, different copies of the same gene, homologs, orthologs, etc. The mean length of transcripts and genes were 1895 bp and 2115 bp, respectively, and the N50 of the transcripts and genes were 2902 and 2936 bp, respectively (Table 2), which were higher than that in *Dendrobium huoshanense*, *Persea Americana*, and *R. glutinosa* [18,71,72]. These results implied that our assembly quality was suitable for subsequent analyses. In the species distribution analysis of unigenes, more than 43.77% of unigenes were matched to *Sesamum indicum* (Figure 2), which is similar to *R. glutinosa*, a plant of Scrophulariaceae; these results implied that they shared the closer genetic relationship, similar chemical substances, and similar biosynthetic pathways. Catalpol, acteoside, and azafrin are three medicinal ingredients in *C*. *grandiflora* Benth; however, their biosynthetic pathway is unexplored.

So far, catalpol biosynthesis containing terpenoid backbone pathway and iridoid pathway has not been fully deciphered due to the deficiency of detailed information on genetic and molecular levels [20]. In 1993, Damtoft found that 8-epi-deoxyloganic acid, bartsioside, and aucubin are intermediates of catalpol biosynthesis by feeding experiments [48]. Then, Jensen et al. confirmed that catalpol is synthesized via decarboxylated iridoids pathway (Route II), which involved 8-epi-iridodial, 8-epi-iridotrial, and 8-epi-deoxyloganic acid [73]. In 2013, the more detailed route II was proposed in *R. glutinosa* and *P. kurrooa* [21,41]. In 2015, the complete catalpol biosynthesis pathway was hypothesized in *P. kurrooa* according to data of the transcriptome mining, gene expression, and picroside content [17]. In our transcriptomes, 368 unigenes were annotated to the catalpol biosynthetic pathway with 60 unigenes upregulated in leaves and 39 unigenes in roots; simultaneously combined with the fact that *F3D* gene was not expressed in roots, we deduced that catalpol biosynthesis was mainly active in leaves. A recent article showed that, in wild *C. grandiflora* Benth, the content of catalpol is far higher in leaves than in stems and roots [74], which also implied that catalpol is mainly synthesized in leaves other than roots. The discovery of rate-limiting enzymes is essential for synthetic biology; therefore, some genes are discussed here. Catalpol biosynthesis begins with the terpenoid backbone pathway, which contains the MEP and MVA pathways. In the MEP pathway, the DXS enzyme is the first and rate-limiting enzyme, and in *A. annua*, among the three AaDXSs, only AaDXS2 might participate in artemisinin biosynthesis [75]. Contrary to *A. annua*, the *DXS*s were more abundant in our transcriptome, which seems that DXS was not a limiting enzyme in *C. grandiflora* Benth. Further studies are needed to clarify which DXS functions in MEP pathway. A recent report showed that plastidial IDI plays an important role in optimizing the ratio between IPP and DMADP as precursors for different downstream isoprenoid pathways while mutation of *IDI1* reduced the content of carotenoids in fruits, flowers, and cotyledons (except mature leaves) [44]. In our transcriptome, there were 28 *IDI* genes with two upregulated in leaves compared with roots, which highlights their importance in terpenoid backbone biosynthesis (Table 4). However, there were no significant differences for the overall expression of *IDI* genes in roots, stems, and leaves in our transcriptome (Figure 9b). What is interesting is that there was only one *MCS* gene in our transcriptome; however, its expression levels in roots, stems, and leaves were all relatively high, which directly denied that *MCS* was a rate-limiting enzyme gene. According to the expression profile, *MCT* may be a rate-limiting enzyme for roots (Figure 9b). In addition, the relative contribution of the MEP and MVA pathways for a specific pathway is a focus scientist paying attention to. In *P. kurroa*, the biosynthesis of picroside-I is contributed solely by the MEP pathway [17]. In *Taxus baccata*, the MEP pathway provides the main source of universal terpenoid precursor IPP [76]. However, in *C. grandiflora* Benth, the contribution of the MEP and MVA pathways for catalpol biosynthesis remains to be clarified and it will be resolved by the inhibition experiments in the future. 

Acteoside biosynthesis was first studies in an *O. europaea* cell with feeding experiments, which outline the basic pathway profile: caffeoyl moiety was synthesized through the phenylalanine-derived pathway including intermediates cinnamic acid, p-coumaric acid, and caffeic acid, while hydroxytyrosol moiety was formed via the tyrosine-derived pathway including two alternative routes [26]. Then, HCT enzyme which connects the caffeoyl moiety and the hydroxytyrosol moiety, UGT enzymes, and the corresponding enzymes of the phenylalanine-derived pathway and tyrosine-derived pathway were hypothesized in *R. glutinosa* [28]. All of the acteoside pathway genes were found in our transcriptome of *C. grandiflora* Benth. Expression profiles showed that genes involved in both the phenylalanine-derived pathway and the tyrosine-derived pathway were more abundant in leaves and stems compared to roots, especially for the *PAL* and *PPO* genes (Figure 10b). This is consistent with the reports that, in *Harpagophytum procumbens*, the content of acteoside was higher in leaves and stems than in roots and that, in *Sesamum indicum*, the content of acteoside in leaves is far higher than in stems and roots [25,77].

Studies have shown that PAL is an entry-point enzyme which can convert *L*-Phe into trans-cinnamic acid and that it plays a vital role in channeling carbon flux from primary metabolism into the phenylpropanoid pathway [78]. So far, *PAL* gene has been cloned from many medicinal plants, such as *Ocimum basilicum* [79], *Ginkgo biloba* [80], *Salvia miltiorrhiza* [81], and *A. annua* [82]. In *G. biloba*, the highest expression of *GbPAL* gene was found in leaves, followed by stems, and the lowest expression was in roots; transcription levels of *GbPAL* were closely related to flavonoid accumulation [80]. In *R. glutinosa*, the *RgPAL* gene (CL1389.Contig1) shared the same expression pattern as in *G. biloba* [28]. In *A. annua*, the highest expression of the *AaPAL* gene was found in young leaves and the lowest expression of that was in roots [82]. In plants, *PAL* gene is a multi-gene family and the gene number ranges from 4 in *A. thaliana* to more than 12 in tomato and potato [83]. For example, there are 6 *PAL* genes in *R. glutinosa* [28]. Recently, three different redundancy phenomena including active compensation in ligand plus passive compensation in receptor in tomato, passive compensation in ligand plus active compensation in receptor in Arabidopsis, and active compensation in both in corn have been figured out [84]; however, which type does the *CgPAL* genes belong to and whether they benefit the plants themselves in *C. grandiflora* Benth remain to be discovered. Unlike potato, the *PAL* gene family is highly redundant but underutilized due to the highly silencing mechanism in tomato [83]. In our transcriptome, there are 19 *PAL* genes and their highest expressions are found in leaves and stems with the lowest expression in roots (Figure 10b), which is similar to that in *G. biloba*, *R. glutinosa*, and *A. annua* [28,80,82]. Our transcriptome profiling data showed that 10 of 19 *CgPAL* genes were not expressed or slightly expressed in roots, stems, and leaves (Appendix A), which implied that gene silencing was also active in *C. grandiflora* Benth, and DNA cytosine methylation may account for this phenomenon [83]. A recent report showed that functional redundancy among *BZR/BEH* (*BRASSINAZOLE-RESISTANT*/*BRI1-EMS-SUPRESSOR1/BRASSINAZOLE-RESISTANT1 HOMOLOG*) gene family members is not necessary for trait robustness [85]. Even in tomato, only *PAL5* was expressed under environmental stimuli [83]. Therefore, *PAL* genes including the 3 significantly upregulated and 1 significantly downregulated in leaf vs. root in *C. grandiflora* Benth played important roles in acteoside biosynthesis (Table 5).

Polyphenol oxidase is usually undesirable in fruit and vegetable due to the browning, while it is desirable in tea, coffee, cocoa, etc. for the pigmentation [86]. Polyphenol oxidase (1,2-benzenediol: oxygen oxidoreductase), also known as tyrosinase, catechol oxidase, and laccase according to the specific substrate and reaction mechanism, is a group of copper-containing proteins [86,87]. A typical PPO protein contains three conservative regions: an N-terminal transit peptide that is responsible for the import of PPO into the thylakoid lumen; a di-copper center, each with three histidine residues to bind a copper atom; and a C-terminal region [88]. Polyphenol oxidases can catalyze two quite different types of reactions: monophenol monooxygenases (E.C. 1.14.18.1) activity and o-diphenol oxidation reactions including catechol oxidases (E.C. 1.10.3.1) and laccases (E.C. 1.10.3.2) activity [87]. In plants, polyphenol oxidase is localized in chloroplasts and the reaction product accumulated in thylakoid [89]. The number of *PPO* gene ranges from 1 to 13 in land plants with 0 for green algae and *A. thaliana*, and tandem duplications of the *PPO* gene family is common in dicotyledon [88]. In our transcriptome, 11 *PPO* genes were clustered into three groups. Expression levels of the upper group including *PPO7*, *PPO9*, *PPO10*, and *PPO11* were higher in leaves and stems compared with roots, while that of the bottom group including *PPO1*, *PPO2*, and *PPO3* were higher in roots and stems than in leaves with the somewhat low expressions in middle group including *PPO4*, *PPO5*, *PPO6*, and *PPO8* (Appendix A). Phylogenetic analysis of 11 CgPPOs with 6 PPOs of *Solanum melongena* and 6 PPOs of *Solanum lycopersicum* showed that all of our CgPPO proteins are clustered into one clade and that the other 12 PPO proteins formed another two clades (Appendix A). These species-specific PPO clades were also found in four major land plant lineages including *Populus trichocarpa*, *Glycine max*, *Vitis vinifera*, and *Aquilegia coerulea*, which implied that *CgPPO* genes were also formed by independent burst of gene duplication [88].

Azafrin (C_27_H_38_O_4_) derivates from tetraterpenoids (C_40_). It has been found in many medicinal plants such as rhizome of *Alectra chitrakutensis*, *Bergenia ciliate*, *Caralluma umbellate*, and *Alectra parasitica*, and it has the functions of being antimicrobial, anti-inflammatory, analgesic, antioxidant, treatment of cardiovascular diseases [90,91,92,93]. Roots of *C. grandiflora* Benth display orange-yellow color, which is largely due to the presence of abundant azafrin as *A. parasitica* [93]. So far, the biosynthetic pathway of azafrin is not established from perspectives of chemistry and biology. There are studies implying that excentric cleavage of carotenoid compounds is a possible route [94]. CCD7 can catalyze β-carotene (C_40_) into 10′-apo-β-carotenal (C_27_) and ionone (C_13_) to support the above hypothesis [39]. In the view of molecular structure, the differences between 10′-apo-β-carotenal and azafrin are one terminal carboxyl group and two hydroxyl groups in cyclohexane skeleton. Therefore, two reactions are indispensable from 10‘-apo-β-carotenal to azafrin: one is to convert the aldehyde group into carboxyl groups, and the other is to insert two oxygen atoms into cyclohexane skeleton to generate two hydroxyl groups. The ALDH superfamily comprises a group of enzymes involved in the NAD^+^ (Nicotinamide Adenine Dinucleotide) or NADP^+^ (Nicotinamide Adenine Dinucleotide Phosphate)-dependent conversion of various aldehydes to their corresponding carboxylic acids [95]. Although there are only 76 NAD^+^-dependent *ALDH* genes in our transcriptome, they are candidate genes for azafrin biosynthesis. In plant, CYP450s are responsible for many oxidative reactions such as hydroxylation, epoxidation, dealkylation, and dehydration, and the reactions catalyzed by CYP450s are irreversible [53]. There are 413 CYP450 unigenes in our transcriptome, of which 5 are significantly upregulated (log_2_(FC) > 10) and 5 are significantly downregulated in leaf vs. root (Table 6). They can be candidate genes of azafrin biosynthesis. The key enzymes determine the flux of the pathway, and the expression of the key enzyme gene dominates the number of enzymes. In marigold, the expression level of the *LCYE* gene in petals and *LCYB* gene in leaves were positively correlated with the lutein content [96]. In *Momordica cochinchinensis*, transcriptional regulation of genes including *HMGR*, *HDS*, *PSY*, *PDS*, *ZDS*, *CRTISO,* and *LCYE* may determine the alteration of carotenoid content during fruit ripening [97]. Our transcriptome data showed that only the expression levels of *HDS*, *PSY*, *ZDS*, and *CRTISO* were more abundant in roots than leaves and stems (Figure 11b). However, trace expression of the *DWARF27* gene in leaves and low expression of the *CCD7* gene in roots, stems, and leaves suggested that they were two rate-limiting enzymes in azafrin biosynthesis. *DWARF27*, which exhibits increased tillers and reduced plant height, was first studied in rice [98]. It encodes an iron-containing protein localized in chloroplasts and is expressed mainly in vascular cells of shoots and roots [98]. Further studies indicated that DWARF27 is an all-trans/9-cis isomerase which can convert all-trans-β-carotene into 9-cis-β-carotene in vivo and in vitro [99]. Obviously, DWARF27 is vital for azafrin biosynthesis. What is interesting is that there is only one *CCD7* gene in our transcriptome which coincides with the all *CCD7* genes identified including maize, rice, sorghum, *Selaginella moellendorfii*, *Physcomitrella patens*, and *Chlamydomonas reinhardtii* and is a single copy [100]. The highest expression of the *CCD7* gene was found in roots among maize, *A. thaliana*, pea, and petunia [100]. However, the highest expression in our transcriptome is in stems.

In the future, studies related to catalpol, acteoside, and azafrin biosynthesis will focus on the following aspects: (1) to construct a transgenic system for *C. grandiflora* Benth according to the successive tissue culture technology for verification of gene function, to characterize the putative genes of three pathways, and to verify their functions by enzyme assays in vitro or to overexpress them in vivo; (2) to explore the correlation between the contents of active component and related gene expression levels, to clone the putative TFs, and to verify their functions in the biosynthesis of active components via chromatin immunoprecipitation and overexpression in vivo; and (3) to figure out the biosynthetic pathway using feeding experiments with suspension cells.

## 4. Materials and Methods

### 4.1. Plant Materials and RNA Isolation

The artificial, cultivated *Centranthera grandiflora* Benth was grown in fields with *Cyperus rotundus* in Yushancheng base, Yuxi Flyingbear Agricultural Development Company Limited, Yuxi, Yunnan Province, China (Figure 15a). Three healthy plants with the same growth potential were selected, and the fresh roots, stems, and leaves were collected from one-year-old *C. grandiflora* plants on May 7, 2018 (Figure 15b). Materials from three individual plants were collected using scissors to yield 1 g of root, stem, and leaf samples (BioSample accessions: SAMN12499651, SAMN12499652, SAMN12499653, SAMN12499654, SAMN12499655, SAMN12499656, SAMN12499657, SAMN12499658, and SAMN12499659). After wrapping with tinfoil and tagging, all samples were immediately frozen in liquid nitrogen and stored at −80 °C.

For total RNA extraction and quality control, refer to Zhang et al. [101].

### 4.2. Library Preparation for Transcriptome Sequencing

A total amount of 1.5 μg RNA per sample was used as input material for the RNA sample preparations. Sequencing libraries were generated using the NEBNext^®^ UItra™ RNA Library Prep Kit for Illumina (NEB, Ipswich, MA, USA) following the manufacturer’s recommendations. Index codes were added to attribute sequences to each sample. Briefly, mRNA was purified from total RNA using poly-T oligo-attached magnetic beads. Fragmentation was carried out using divalent cations under elevated temperature in a NEBNext First Strand Synthesis Reaction Buffer (5×). First strand cDNA was synthesized using random hexamer primer and M-MuLV Reverse Transcriptase (RNase H^−^). Second-strand cDNA synthesis was subsequently performed using DNA Polymerase I and RNase H. Remaining overhangs were converted into blunt ends via exonuclease/polymerase activities. After adenylation of 3′ ends of DNA fragments, NEBNext Adaptor with hairpin loop structure was ligated to prepare for hybridization. In order to select cDNA fragments preferentially 250–300 bp in length, the library fragments were purified with AMPure XP system (Beckman Coulter, Beverly, USA). Then, 3 μL USER Enzyme (NEB, Ipswich, MA, USA) was used with size-selected, adaptor-ligated cDNA at 37 °C for 15 min followed by 5 min at 95 °C before PCR. PCR was performed with Phusion High-Fidelity DNA polymerase, Universal PCR primers, and Index(X) Primer. At last, PCR products were purified (AMPure XP system) and library quality was assessed on the Agilent Bioanalyzer 2100 system (Agilent Technologies, Palo Alto, CA, USA).

### 4.3. Clustering and Sequencing

The clustering of the index-coded samples was performed on a cBot Cluster Generation System using TruSeq PE Cluster Kit v3-cBot-HS (Illumia, San Diego, CA, USA) according to the manufacturer’s instructions. After cluster generation, the library preparations were sequenced on an Illumina Hiseq2000 platform and paired-end reads were generated.

### 4.4. Data Filtering and Transcriptome Assembly

The flow of bioinformatics analysis is showed in Figure 16. Before assembly, raw reads containing adaptors, more than 10% N bases, and more than 50% of low quality were removed to obtain clean reads. Meanwhile, Q20, Q30, and GC content were used to assess the data quality. All the subsequent analyses were based on these clean reads. As there are no reference genomes available for *C. grandiflora*, the clean reads of roots, stems, and leaves were assembled together. The paired-end reads of each sample were merged into one interleaved fastq file. All the nine pooled files were assembled using Trinity software (version: r20140413p1) (Cambridge, MA, USA) with min_k-mer_cov set to 2 and all other parameters settings as default [102]. After clustering and de-redundancy by Corset software (version: 1.07) (VIC, Austrilia) [103], the clean nonredundant unigenes was generated.

### 4.5. Gene Functional Annotation

Gene function was annotated based on the following databases: Nr (NCBI nonredundant protein sequences, diamond v0.8.22, *e*-value = 10^−5^), NT (NCBI nucleotide sequences, NCBI blast 2.2.28+, *e*-value = 10^−5^), PFAM (Protein family, HMMER 3.0 package, hmmscan *e*-value = 10^−2^), SwissProt (a manually annotated and reviewed protein sequence database, diamond v0.8.22, *e*-value = 10^−5^), KOG/COG (Clusters of Orthologous Groups of proteins/euKaryotic Ortholog Groups, diamond v0.8.22, *e*-value = 10^−5^), KAAS (version: r140224, *e*-value = 10^−10^), and GO (Blast2GO v2.5 and inhouse script, *e*-value = 10^−6^). To figure out the TF families involved in the active ingredient biosynthesis, iTAK software (https://github.com/kentn/iTAK/) was used to predict the TF. Its basic principle is to identify TF by hmmscan using TF family and rules defined by classification in the database. For the identification and classification methods of TF, refer to Perez-Rodriguez et al. [104].

### 4.6. Differential Expression Analysis

The calculation of unigene expression was performed using the RPKM method, and gene expression levels were estimated by RSEM (version: 1.2.15, parameter for bowtie2: mismatch 0) for each sample [43]. Differential expression analysis of two organs was performed using the DESeq R package 1.10.1. DESeq provides statistical routines for determining differential expression in digital gene expression data using a model based on the negative binomial distribution. The resulting *p* values were adjusted using Benjamini and Hocberg’s approach for controlling false discovery rates. Genes with a threshold of foldchange ≥ 2 and *p*-value < 0.05 found by DESeq were assigned as differentially expressed. 

### 4.7. GO Enrichment and KEGG Pathway Enrichment Analysis

Gene ontology enrichment analysis of the differentially expressed genes (DEGs) was implemented by the GOseq (version: 1.10.0) and topGO (version: 2.10.0) R packages based Wallenius non-central hyper-genometric distribution, which can adjust for gene length bias in DEGs [105]. KOBAS software (Beijing, China) was used to test the statistical enrichment of DEGs in KEGG pathways [106]. The expressed genes (FPKM ≥ 1) were used as background with a corrected *p*-value ≤ 0.05 for both enrichment analyses.

### 4.8. qRT-PCR Analysis

Total RNA was extracted from roots, stems, and leaves of annual *C. grandiflora* Benth. The first strand of DNA was synthesized using reverse transcription kit PrimeScript RT Master Mix (Perfect Real Time) (Takara, Dalian, China). Specific primers were designed according to the selected gene sequences for expression analysis (Appendix A). Using the *C. grandiflora* Benth *CgUbi* gene (Accession number: MK256646) as an internal reference, qPCR was performed using chimeric fluorescence detection kit TB Green Premix Ex Taq II (Takara, China). Each reaction was repeated three times. Reaction was amplified by LightCycler 480II fluorescent quantitative PCR (Roche, Basel, Switzerland). After the amplification, results were calibrated by internal reference gene and the relative gene expressions in roots, stems, and leaves were calculated automatically by the 2^−ΔΔ*C*t^ method.

### 4.9. Data Submission

This Transcriptome Shotgun Assembly project (PRJNA558809) has been deposited at DDBJ/ENA/GenBank under the accession GHUX00000000. The version described in this paper is the first version, GHUX01000000.

## 5. Conclusions

*Centranthera grandiflora* Benth has been used to prevent and treat CVDs for a long time; however, the biosynthesis pathway of its active components including catalpol, acteoside, and azafrin remains undeciphered. Transcriptome sequencing technology is an effective way to discover the genes of this herb’s biosynthesis pathways and its regulatory mechanisms. In this study, nine cDNA libraries were constructed from the roots, stems, and leaves of *C. grandiflora* Benth and sequenced by an Illumina Hiseq2000 platform. As a result, 438,112,930 clean reads were obtained and 153,198 unigenes were assembled. Among these genes, 557, 213, and 161 unigenes were annotated into catalpol, acteoside, and azafrin biosynthetic pathways, respectively. Azafrin can be synthesized through β-carotene, 9-cis-β-carotene, and 10′-apo-β-carotenal with the corresponding enzymes DWARF27, CCD7, ALDH, and CYP450. Also, the *PAL* gene silencing phenomenon is discovered and discussed. The candidate TF MYBs involved in the regulation of these pathways were proposed. Our results represent the first genomic resource for *C. grandiflora* Benth, which is a starting point for exploration of this valuable herb in molecular biology.

## Figures and Tables

**Figure 1 ijms-20-06034-f001:**
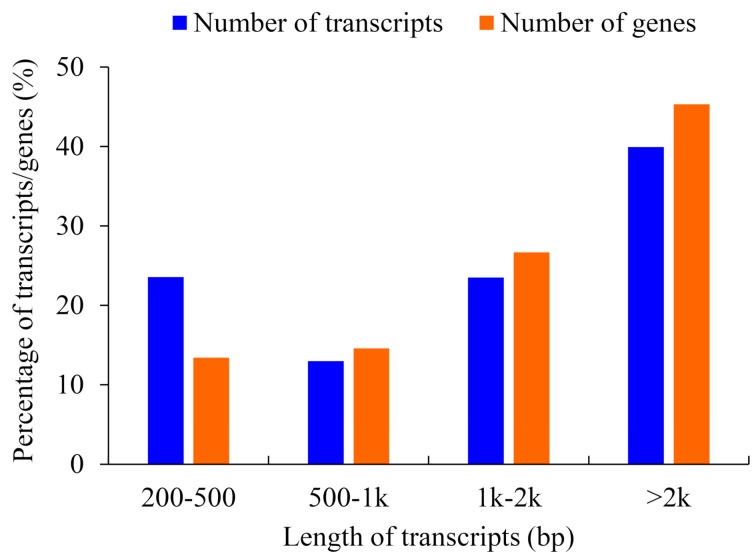
Length distribution frequency of spliced transcripts and deduced genes.

**Figure 2 ijms-20-06034-f002:**
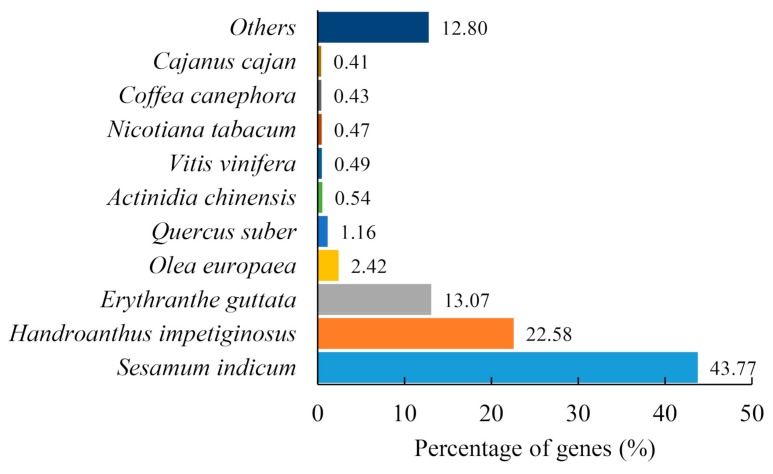
Species distribution of top 10 BLASTx hits against the NR database.

**Figure 3 ijms-20-06034-f003:**
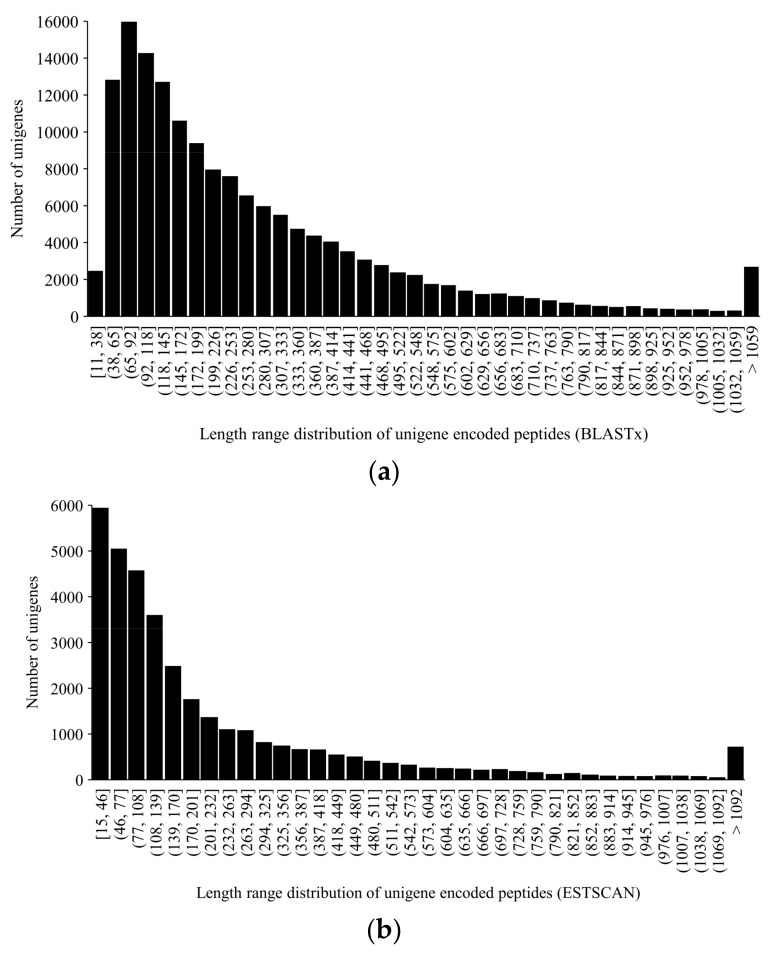
Length range distributions of unigene encoded peptides: (**a**) Peptides predicted by BLASTx searches against NR and Swissprot databases; (**b**) peptides predicted by software ESTSCAN 3.0.3.

**Figure 4 ijms-20-06034-f004:**
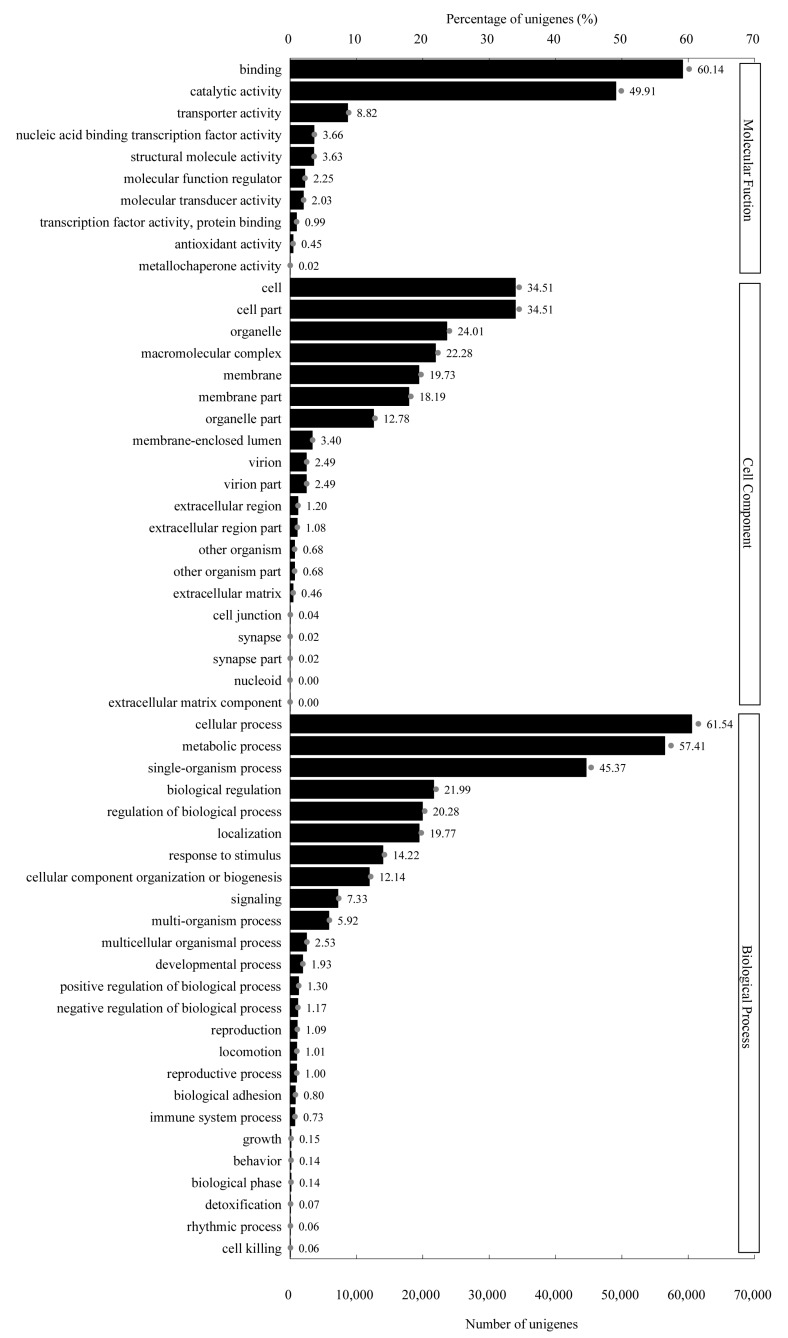
GO classification map: The ordinate represents the next-level GO term of the three GO categories, while the abscissa represents the number of genes annotated into the corresponding term and its proportion of the total number of annotated genes. Three basic categories of GO term, from top to bottom, are the molecular function, cell components, and biological processes.

**Figure 5 ijms-20-06034-f005:**
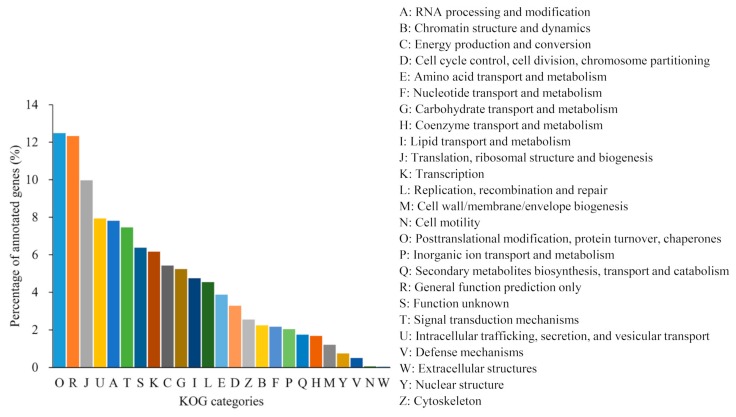
KOG classification map: The abscissa represents KOG groups, while the vertical axis represents the percentage of annotated genes.

**Figure 6 ijms-20-06034-f006:**
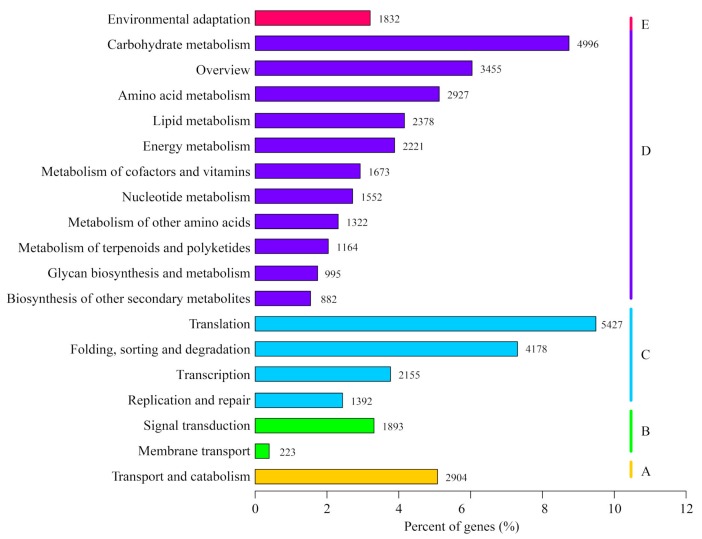
KEGG classification map: The ordinate is the pathway, and the abscissa is the proportion of genes belonging to the corresponding pathway. These genes were divided into five categories: **A**. Cellular Processes; **B**. Environmental Information Processing; **C**. Genetic Information Processing; **D**. Metabolism; and **E**. Organismal Systems.

**Figure 7 ijms-20-06034-f007:**
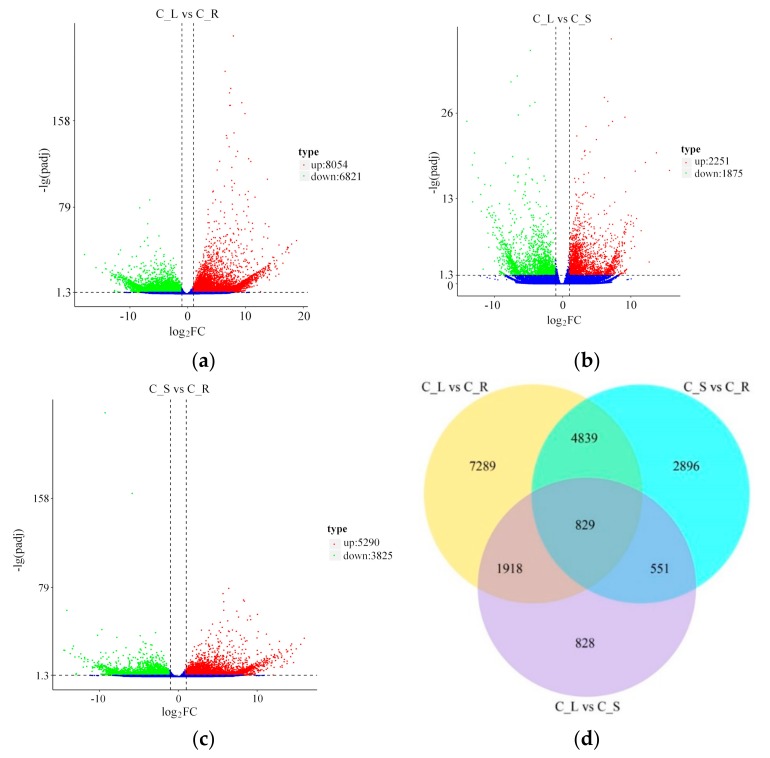
Differentially expressed genes (DEGs) in different comparisons. Volcano plots of the DEGs in different comparisons: The red dots mean significantly upregulated genes, and the green dots represent significantly downregulated genes. The black dots represent non-DEGs. (**a**) C_L vs. C_R volcano; (**b**) C_L vs. C_S volcano; and (**c**) C_S vs. C_R volcano. (**d**) Venn diagram of DEGs in different comparisons: All DEGs are clustered into three comparison groups represented by three circles. Overlapping parts of the different circles represent the number of DEGs in common among those comparison groups.

**Figure 8 ijms-20-06034-f008:**
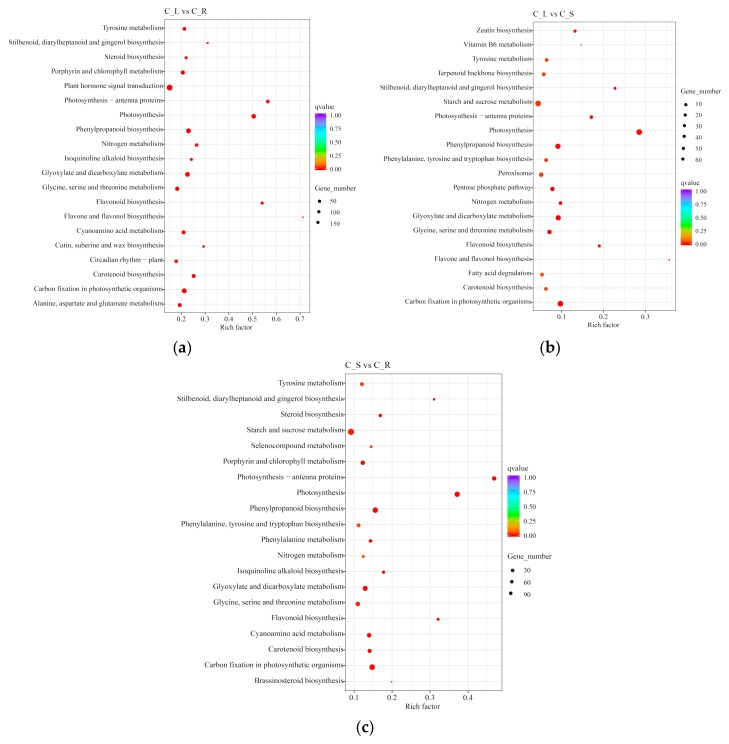
Top 20 of KEGG pathway enrichment of DEGs: The y-axis indicates the pathway name, and the x-axis indicates the enrichment factor corresponding to the pathway. The q-value is represented by the color of the dot. The number of DEGs is represented by the size of the dots. (**a**) C_L vs. C_R; (**b**) C_L vs. C_S; and (**c**) C_S vs. C_R.

**Figure 9 ijms-20-06034-f009:**
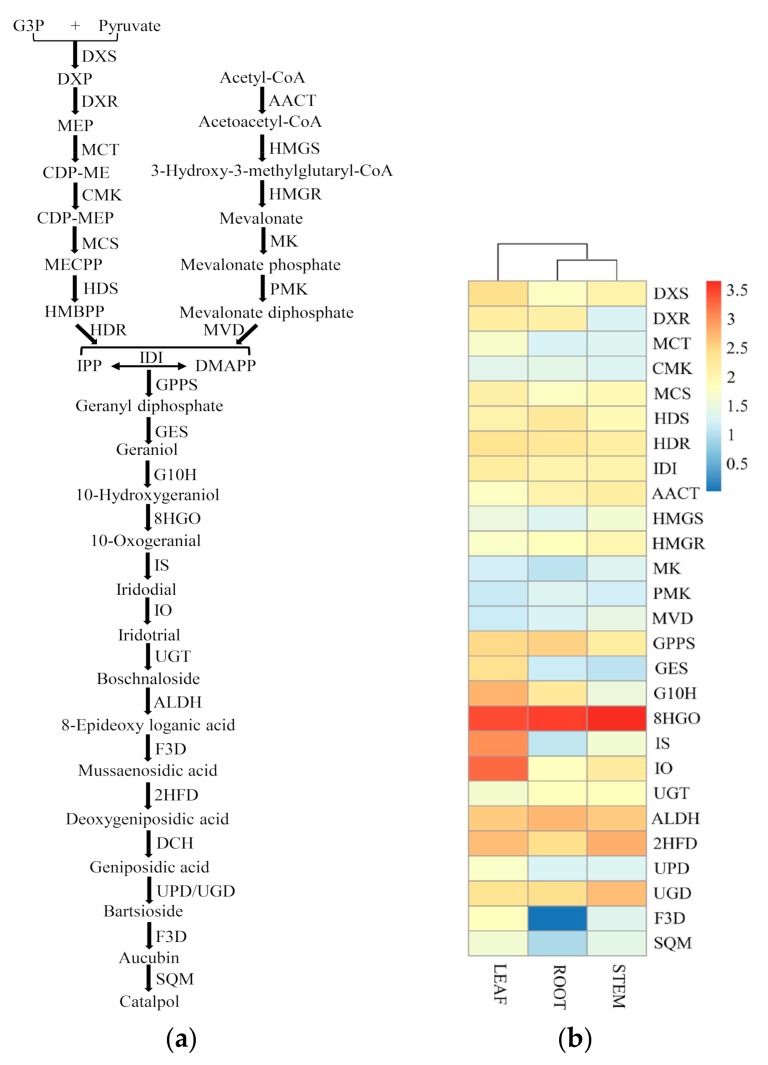
Expression of unigenes in the putative pathway of terpenoid backbone and catalpol biosynthesis in *C. grandiflora* Benth: (**a**) Proposed biosynthetic pathway of terpenoid backbone and catalpol. (**b**) Heatmap based on the expression level of unigenes involved in terpenoid backbone and catalpol biosynthesis across three tissues in *C. grandiflora* Benth. The expression level is the sum of all the unigenes for each gene, and log_10_(sum(FPKM)+1) was used to plot the heatmap. Candidate unigenes were selected according to the annotation. Abbreviations: G3P, Glyceraldehyde 3-phosphate; DXP, 1-deoxy-D-xylulose-5-phosphate; MEP, 2-C-methyl-D-erythritol 4-phosphate; CDP-ME, 4-(Cytidine 5′-diphospho)-2-C-methyl-D-erythritol; MECPP, 2-C-methyl-D-erythritol-2,4-cyclodiphosphate; HMBPP, 1-hydroxy-2-methyl-2-butenyl 4-diphosphate; IPP, isopentenyl pyrophosphate; DMAPP, dimethylallyl pyrophosphate.

**Figure 10 ijms-20-06034-f010:**
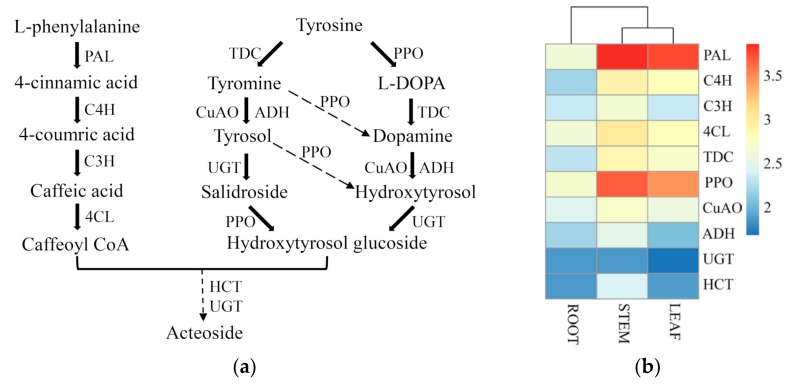
Expression of unigenes in putative pathways of acteoside biosynthesis in *C. grandiflora* Benth: (**a**) Proposed biosynthetic pathway of acteoside. The solid arrow represents the known steps, and the dashed arrows denote the putative steps. (**b**) Heatmap based on the expression level of unigenes involved in acteoside biosynthesis across three tissues: Candidate unigenes were selected according to the annotation.

**Figure 11 ijms-20-06034-f011:**
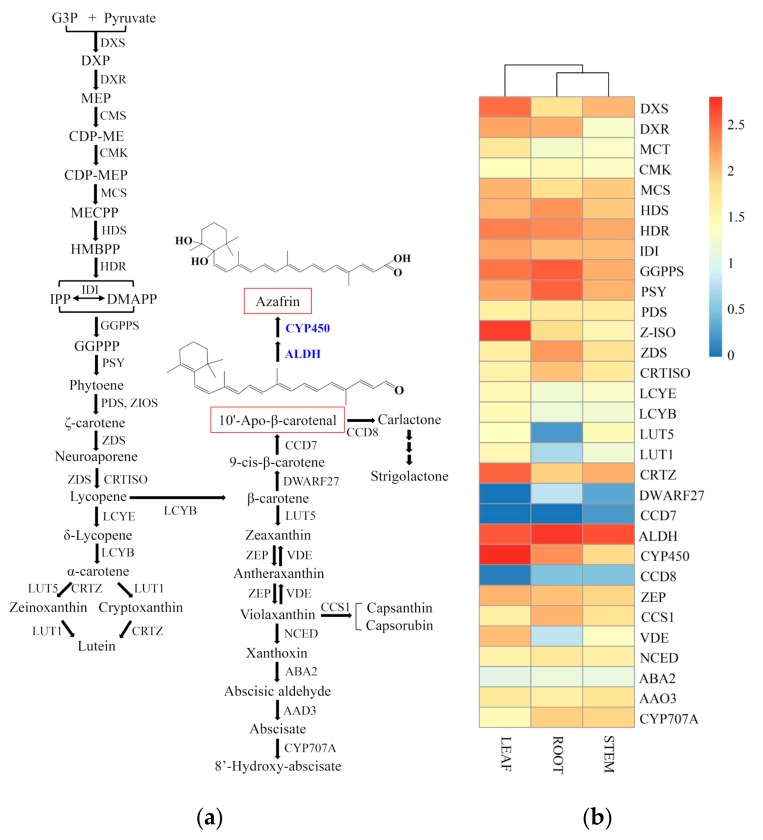
The putative carotenoid biosynthesis pathway and the heatmap of corresponding genes in *C. grandiflora* Benth: (**a**) Proposed biosynthetic pathway of carotenoid. Here, the hypothesis that 10′-apo-β-carotenal can be converted into azafrin by ALDH and CYP450 is proposed. (**b**) Heatmap based on the expression level of unigenes involved in carotenoid biosynthesis across three tissues. For CYP450, only genes of Log_2_(FC) > 10 (leaf vs. root) were selected for map. Candidate unigenes were selected according to the annotation.

**Figure 12 ijms-20-06034-f012:**
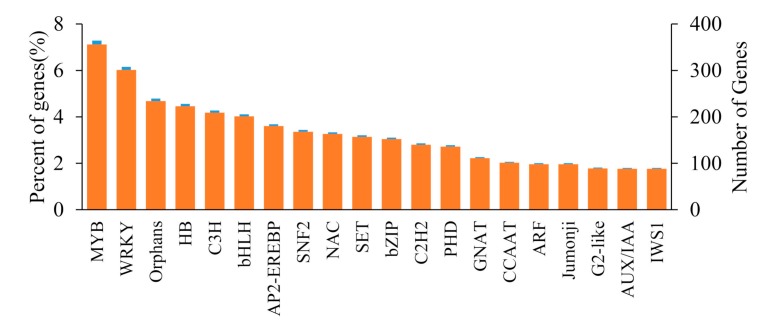
Top 20 transcription factors in *C. grandiflora* Benth.

**Figure 13 ijms-20-06034-f013:**
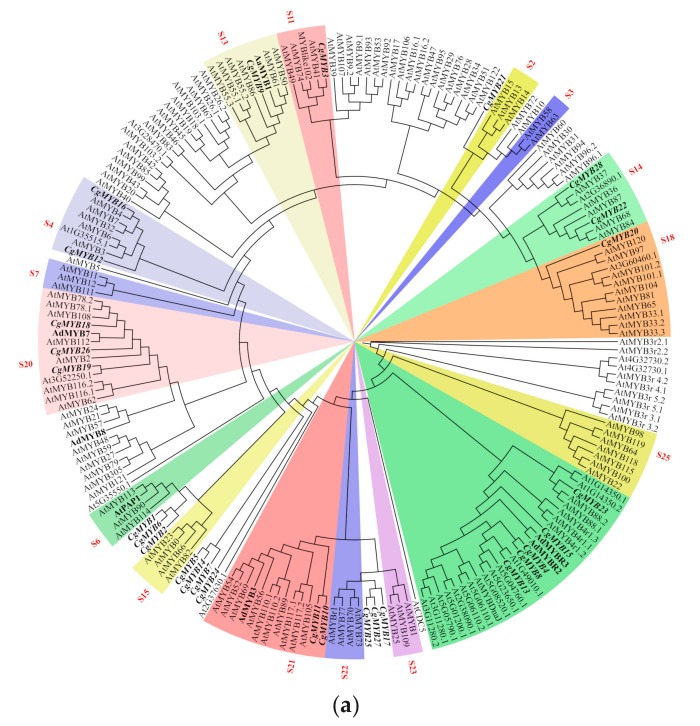
Phylogenetic analysis and expression level of MYBs from *C. grandiflora* Benth: (**a**) Phylogenetic analysis of CgMYBs. Amino acid sequences were aligned using the ClustalX2 program, and evolutionary distances were calculated using phyML software with the maximum likelihood statistical method. The sequences of *C. grandiflora* Benth are listed in Appendix A. The sequences of *Arabidopsis thaliana* come from PLANTTFDB (https://planttfdb.cbi.pku.edu.cn), while that of *Artemisia annua* and *Actinidia deliciosa* come from NCBI (https://www.ncbi.nlm.nih.gov). (**b**) Expression level of CgMYBs: Expression level for each gene is represented by the average RPKM in roots, stems, and leaves.

**Figure 14 ijms-20-06034-f014:**
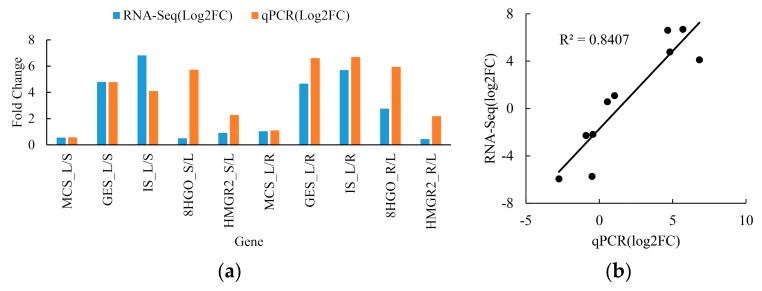
The correlation analysis of gene expression pattern between RNA-Seq and qPCR in roots, stems, and leaves in *C. grandiflora* Benth. (**a**) Expression patterns of *MCS*, *GES*, *IS*, *8HGO* and *HMGR2* gene by RNA-Seq and qPCR; (**b**) Correlation analysis of gene expression between RNA-Seq and qPCR. Each qPCR was biologically repeated three times.

**Figure 15 ijms-20-06034-f015:**
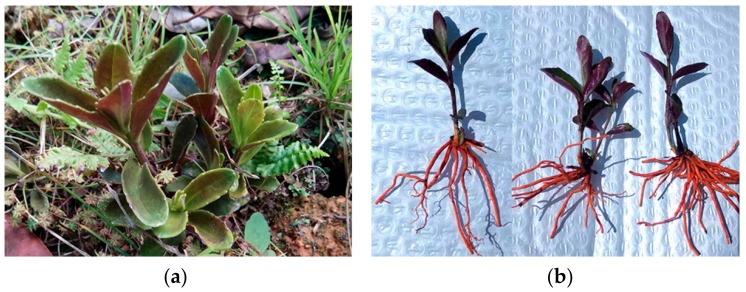
Plant materials of *C. grandiflora* Benth: (**a**) *C. grandiflora* Benth growing in the field with *Cyperus rotundus* and (**b**) Roots, stems, and leaves used in experiments for sequencing and qRT-PCR.

**Figure 16 ijms-20-06034-f016:**
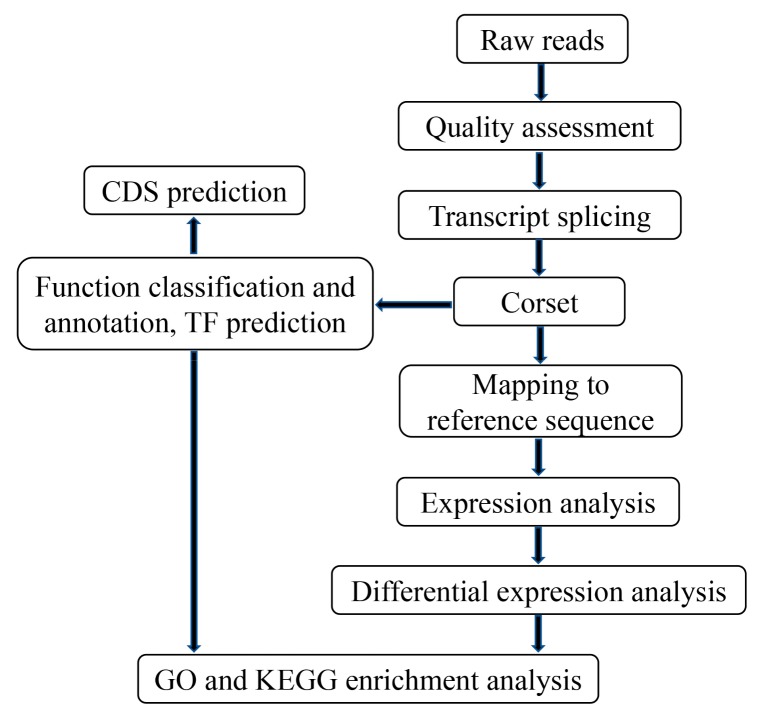
Flow chart of transcriptome bioinformatics analysis for *C. grandiflora* Benth.

**Table 1 ijms-20-06034-t001:** Quality assessment of the sequencing data.

Sample	Raw Reads	Clean Reads	Clean Bases	Error (%)	Q20 (%)	Q30 (%)	GC (%)
C_R1	49187308	48514118	7.28G	0.03	95.57	89.01	47.36
C_R2	49633082	48674546	7.3G	0.03	97.80	93.85	49.17
C_R3	48837996	47783570	7.17G	0.03	97.94	94.13	48.63
C_S1	55750138	55002348	8.25G	0.03	95.53	89.08	47.82
C_S2	49324046	48163930	7.22G	0.03	97.47	93.33	49.39
C_S3	50001598	48875538	7.33G	0.03	97.63	93.61	49.65
C_L1	50405504	49123820	7.37G	0.03	95.66	89.19	48.69
C_L2	45678578	45075310	6.76G	0.03	97.59	93.45	49.89
C_L3	47606624	46899750	7.03G	0.03	94.99	88.32	49.60

Note: C_R1, C_R2, and C_R3: three root samples; C_S1, C_S2, and C_S3: three stem samples; and C_L1, C_L2, and C_L3: three leaf samples.

**Table 2 ijms-20-06034-t002:** Length frequency distribution of the spliced transcripts and genes.

	Min Length	Mean Length	Median Length	Max Length	N50	N90	Total Nucleotides
Transcripts	201	1895	1574	16,816	2902	1089	329,518,919
Genes	201	2115	1824	16,816	2936	1195	323,991,974

**Table 3 ijms-20-06034-t003:** Statistical results of gene annotation.

Item	Number of Unigenes (n)	Percentage (%)
Annotated in NR	127,767	83.39
Annotated in NT	96,216	62.80
Annotated in SwissProt	103,257	67.40
Annotated in PFAM	98,364	64.20
Annotated in GO	98,364	64.20
Annotated in KOG	44,170	28.83
Annotated in KEGG	57,190	37.33
Annotated in all Databases	26,652	17.39
Annotated in at least one Database	132,896	86.74
Total unigenes	153,198	100

Note: NR (nonredundant protein sequences), NT (Nucleotide collection), PFAM (Protein family), GO (Gene Ontology), KOG (euKaryotic Orthologous Groups), KEGG (Kyoto Encyclopedia of Genes and Genomes).

**Table 4 ijms-20-06034-t004:** Putative genes of the mevalonate (MVA), 2-C-methyl-D-erythritol-4-phosphate (MEP), and catalpol biosynthesis pathways.

Pathway	Gene	Gene Name	EC	Number	Upregulated (log_2_(FC) > 1, L vs. R)	Downregulated (log_2_(FC) > 1, L vs. R)
MVA	*AACT*	acetyl-CoA C-acetyltransferase	2.3.1.9	24		2
	*HMGS*	hydroxymethylglutaryl-CoA synthase	2.3.3.10	8	2	
	*HMGR*	hydroxymethylglutaryl-CoA reductase	1.1.1.34	9		
	*MK*	mevalonate kinase	2.7.1.36	6	1	
	*PMK*	phosphomevalonate kinase	2.7.4.2	21		3
	*MVD*	diphosphomevalonate decarboxylase	4.1.1.33	6		
MEP	*DXS*	1-deoxy-D-xylulose-5-phosphate synthase	2.2.1.7	67	12	
	*DXR*	1-deoxy-D-xylulose-5-phosphate reductoisomerase	1.1.1.267	10		
	*MCT*	2-C-methyl-D-erythritol 4-phosphate cytidylyltransferase	2.7.7.60	3	1	
	*CMK*	4-diphosphocytidyl-2-C-methyl-D-erythritol kinase	2.7.1.148	24		
	*MCS*	2-C-methyl-D-erythritol 2,4-cyclodiphosphate synthase	4.6.1.12	1	1	
	*HDS*	(*E*)-4-hydroxy-3-methylbut-2-enyl-diphosphate synthase	1.17.7.1/1.17.7.3	21		
	*HDR*	4-hydroxy-3-methylbut-2-en-1-yl diphosphate reductase	1.17.7.4	11		
	*IDI*	isopentenyl pyrophosphate isomerase	5.3.3.2	28	2	
Catalpol	*GPPS*	geranyl diphosphate synthase	2.5.1.1	32	5	6
	*GES*	geraniol synthase	3.1.7.11	9	3	
	*G10H*	geraniol 10-hydroxylase	1.14.13.152	14	8	2
	*8HGO*	8-hydroxygeraniol oxidoreductase	1.1.1.324	53	13	11
	*IS*	iridoid synthase	1.3.1.99	5	3	
	*IO*	iridoid oxidase		44	3	2
	*UGT*	UDP-glucosyl transferase	2.4.1.	22	4	4
	*ALDH*	aldehyde dehydrogenase	1.2.1.3	76	7	10
	*F3D*	flavanone 3-dioxygenase	1.14.11.9	2	1	
	*2HFD*	2-hydroxyisoflavanone dehydratase	4.2.1.105	10	2	
	*UPD*	uroporphyrinogen decarboxylase	4.1.1.37	23	3	
	*UGD*	UDP-glucuronic acid decarboxylase	4.1.1.35	70	3	4
	*SQM*	squalene monooxygenase	1.14.13.132	8	5	

Note: FC represents fold change.

**Table 5 ijms-20-06034-t005:** Putative genes of acteoside biosynthesis pathways.

Gene	Gene name	EC	Number	Upregulated (log_2_(FC) > 1, L vs. R)	Downregulated (log_2_(FC) > 1, L vs. R)
*PAL*	phenylalanine ammonia-lyase	4.3.1.24	19	4	1
*C4H*	cinnamate-4-hydroxylase	1.14.14.91	12	5	
*C3H*	coumarate-3-hydroxylase	1.14.14.96	10	3	1
*4CL*	4-coumarate-CoA ligase	6.2.1.12	35	9	
*TDC*	tyrosine decarboxylase	4.1.1.25	19	3	
*CuAO*	copper-containing amine oxidase	1.4.3.21	53	4	2
*ADH*	alcohol dehydrogenase	1.1.1.1	26	3	3
*UGT*	UDP-glucose glucosyltransferase	2.4.1.35	22	4	4
*PPO*	polyphenol oxidase	1.14.18.1	11	4	5
*HCT*	Shikimate O-hydroxycinnamoyltransferase	2.3.1.133	6	1	

**Table 6 ijms-20-06034-t006:** Putative genes of carotenoid biosynthesis pathways.

Gene	Gene Name	EC	Number	Upregulated (log_2_(FC) > 1, L vs. R)	Downregulated (log_2_(FC) > 1, L vs. R)
*GGPPS*	geranylgeranyl diphosphate synthase	2.5.1.29	15	3	6
*PSY*	phytoene synthase	2.5.1.32	9	1	2
*PDS*	phytoene desaturase	1.3.5.5	7	2	1
*Z-ISO*	zeta-carotene isomerase	5.2.1.12	3		2
*ZDS*	zeta-carotene desaturase	1.3.5.6	26	1	4
*crtISO*	carotenoid isomerase	5.2.1.13	10		2
*LCYE*	lycopene epsilon-cyclase	5.5.1.18	49	3	1
*LCYB*	lycopene beta-cyclase	5.5.1.19	2	1	
*LUT5*	beta-ring hydroxylase	1.14.-.-	18	8	
*LUT1*	carotenoid epsilon hydroxylase	1.14.99.45	14	2	
*CRTZ*	beta-carotene 3-hydroxylase	1.14.13.129	11	6	5
*DWARF27*	beta-carotene isomerase	5.2.1.14	2		1
*CCD7*	9-cis-beta-carotene 9′,10′-cleaving dioxygenase	1.13.11.68	1		
*ALDH*	aldehyde dehydrogenase	1.2.1.3	76	7	10
*CYP450*	cytochrome P450		10	5	5
*CCD8*	carlactone synthase	1.13.11.69	11		
*ZEP*	zeaxanthin epoxidase	1.14.15.21	17	1	4
*CCS1*	capsanthin/capsorubin synthase	5.3.99.8	4		1
*VDE*	violaxanthin de-epoxidase	1.23.5.1	12	3	
*NCED*	9-cis-epoxycarotenoid dioxygenase	1.13.11.51	33	6	7
*ABA2*	xanthoxin dehydrogenase	1.1.1.288	5	1	
*AAD3*	abscisic-aldehyde oxidase	1.2.3.14	11		
*CYP707A*	(+)-abscisic acid 8′-hydroxylase	1.14.14.137	10	4	6

**Table 7 ijms-20-06034-t007:** Summary of transcription factor unigenes of *C. grandiflora* Benth.

TF Family	Number of Genes Detected	UpRegulated in Leaves (log_2_(FC) > 2)	Upregulated in Roots (log_2_(FC) > 2)
MYB	356	25	21
WRKY	301	16	44
Orphans	234	14	11
HB	223	25	4
C3H	209	6	7
bHLH	201	21	9
ERF	180	9	21
bZIP	152	5	15
Total	1856	121	132

Note: TF (Transcription Factor), MYB (avian myeloblastosis viral oncogene homolog), WRKY (WRKY domain-containing protein), HB(homeobox), C3H(Cys3His zinc finger domain-containg protein), bHLH (basic Helix-Loop-Helix), ERF(Ethylene Responsive Factor), bZIP (basic region-leucine zipper).

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
