# Peer review of "Analysis of Centranthera grandiflora Benth Transcriptome Explores Genes of Catalpol, Acteoside and Azafrin Biosynthesis"

_ijms, 2019, doi:10.3390/ijms20236034_

Round 1

Reviewer 1 Report

This work performed RNA-seq and de novo transcript assembly of C. grandiflora benth to examine biosynthesis pathways for catalpol, acteoside and azafrin. This manuscript is logically constructed, and data analysis and presentation of results are acceptable. I suggest some minor corrections before publication.

Page numbers behind reference numbers throughout the text should be deleted. For example, “[1] (p. 510-511)” in line 37.

Line 29. “This work is the first transcriptome” is strange. “This work is the first transcriptome analysis” is better.

Figure 8. I cannot read pathway names due to too small size. Please enlarge them.

Lines 526 and 531. “silence” is not common. “silencing” is common.

Typo. Line 525. “duo”. “due” is correct.

Author Response

Response to Reviewer 1 Comments

Point 1: Page numbers behind reference numbers throughout the text should be deleted. For example, “[1] (p. 510-511)” in line 37.

Response 1: The page numbers has been deleted.

Point 2: Line 29. “This work is the first transcriptome” is strange. “This work is the first transcriptome analysis” is better.

Response 2: It has been modified to “This work is the first transcriptome analysis”.

Point 3: Figure 8. I cannot read pathway names due to too small size. Please enlarge them.

Response 3: It has been replaced by high quality figures.

Point 4: Lines 526 and 531. “silence” is not common. “silencing” is common.

Response 4: “silence” has been replaced by “silencing”.

Point 5: Typo. Line 525. “duo”. “due” is correct. 

Response 5: “duo” has been replaced by “due”.

Reviewer 2 Report

The manuscript appears not well curated. A number of weak points are present, I strongly suggest a careful reading and rewriting of each paragraph in order to improve the quality and relevance of the content. I report in the following points some of the weak points I noted.

Lines 144-149: the Authors report about homology. Their use of this term is misleading, they should refer to "sequence similarity" and indicates what threshold in percentage of similarity is applied to consider "high" the similarity.

Fig. 3a has wrong X-axis legend. What is the bar that includes peptides of 38 amino acids? The first oen or the second one?
Similarly, the same error occurs in Fig. 3b.

Line 170-171: "As shown in Figure 4, assignments which fell under 170 biological process ranked the highest, followed by cellular component and molecular function."
What it means?
Lines 171-176: very generic comment. The Authors should comment the biological meaning of thes numbers. Do they find extraordinary/expected/ these values ?
Line 177: do the Authors consider extraordinary/expected that cellular metabolism was active ?
Absolute numbers of occurrence of annotations do not assure significativity of the result, because that annotations could be very frequent. Therefore, a critical evalution should be given, as for example a comparison with other species.

Lines 183-190: as for the previous comment.
Lines 194-204:as for the previous comment.

The Authors must report reference for the Short Read Archive where their results are deposited.

Supplementary Materials apper as a confuse amont of material. I do not find a clear description of what is reported.

Reviewer 3 Report

Review Comments

In this manuscript, the authors present sequence and transcript abundance data for the root, stem and leaf transcriptome of C. grandiflora benth obtained by the Illumina Hiseq2000. More than 438 million clean reads were obtained from root, stem and leaf libraries, which produced 153,198 unigenes. Based on databases annotation, a total of 557, 213 and 161 unigenes were annotated to catalpol, acteoside and azafrin biosynthetic pathway separately. Differentially expressed genes analysis identified 14,875 unigenes differentially enriched between leaf and root with 8,054 upregulated genes and 6,821 downregulated genes. Candidate MYB transcription factors involved in catalpol, acteoside and azafrin biosynthesis were also predicated. This work is the first transcriptome in C. grandiflora benth which will aid the deciphering of biosynthesis pathways and regulatory mechanisms of active components. 

In general, the manuscript has a sort of novelty and is well-written, however the following revisions should be considered:

- Introduction

The introduction should discuss the hypthesis of this work in details.

Additionally, recently related papers should be linked to this work.

The authors should do English language editing as there are some grammar errors.

- Results and discussion

The results are presented in the form of tables and figures. However, there are some figures with very low quality and can not be read or followed, such as Figures 3, 7 and 8. Please replace those figures with high-quality ones and describes them in detail.

The discussion is somewhat well-written but should be linked with recently published work related to this subject.

Lines 479 -503,  this section should be re-written with regard to the significance of this data and its relatedness to the recent. Works

Lines  525-537, this paragraph should also be linked with recent works and explains the hypothesis of this work with regard to these obtained results.

The conclusion section should be written with more adjustments with regard to the significant results obtained in this study.

- Methods

The methods are written briefly. The methods should include how the experiments were designed and analyzed so that the results can be reproducible.

Additionally, bioinformatics analysis should be explained including the software packages used

-Bibliography

Recent references related to the hypothesis and literature should be included. And please discard the old references.
